# Variance based weighting of multisensory head rotation signals for verticality perception

Christopher J. Dakin[1]*[¤], Prateek Kumar[1], Patrick A. Forbes[2], Amy Peters[1], Brian L. Day[1]

**1** Institute of Neurology, University College London, London, England, **2** Department of Neuroscience, Erasmus University Medical Centre, Rotterdam, The Netherlands

¤ Current address: Department of Kinesiology and Health Sciences, Utah State University, Logan, Utah, United States of America

* chris.dakin@usu.edu

**Data Availability Statement:** All relevant data are on Mendeley and are available at: https://data.mendeley.com/datasets/vpngm4j23t/2.

**Funding:** CJD received financial support from a Canadian Institutes for Health Research Post-

## Abstract

We tested the hypothesis that the brain uses a variance-based weighting of multisensory cues to estimate head rotation to perceive which way is up. The hypothesis predicts that the known bias in perceived vertical, which occurs when the visual environment is rotated in a vertical-plane, will be reduced by the addition of visual noise. Ten healthy participants sat head-fixed in front of a vertical screen presenting an annulus filled with coloured dots, which could rotate clockwise or counter-clockwise at six angular velocities (1, 2, 4, 6, 8, 16˚/s) and with six levels of noise (0, 25, 50, 60, 75, 80%). Participants were required to keep a central bar vertical by rotating a hand-held dial. Continuous adjustments of the bar were required to counteract low-amplitude low-frequency noise that was added to the bar's angular position. During visual rotation, the bias in verticality perception increased over time to reach an asymptotic value. Increases in visual rotation velocity significantly increased this bias, while the addition of visual noise significantly reduced it, but did not affect perception of visual rotation velocity. The biasing phenomena were reproduced by a model that uses a multisensory variance-weighted estimate of head rotation velocity combined with a gravito-inertial acceleration signal (GIA) from the vestibular otoliths. The time-dependent asymptotic behaviour depends on internal feedback loops that act to pull the brain's estimate of gravity direction towards the GIA signal. The model's prediction of our experimental data furthers our understanding of the neural processes underlying human verticality perception.

## Introduction

The brain uses multisensory information to estimate the direction of gravity, in part because sensory organs that respond to linear acceleration, e.g. otoliths, also signal body orientation in the gravitational field due to Einstein's equivalence principle [1]. Because of this ambiguity, the brain must use additional sensory information to separate the components of the signal due to each of these two physical stimuli. The prevailing view is that during passive rotation head motion signals are summed over time to estimate changes of head orientation in space, thus permitting the disambiguation of head tilt from linear acceleration [2–15]. A primary source of such head rotation information arises from the semicircular canals, and it has been

Doctoral Fellowship (http://www.cihr-irsc.gc.ca/e/
193.html) and PAF received financial support from
the Netherlands Organization for Scientific
Research: NWO #016. Veni. 188.049 (https://www.
nwo.nl/en). The funders had no role in study
design, data collection and analysis, decision to
publish, or preparation of the manuscript

**Competing interests:** The authors have declared
that no competing interests exist.

shown that canal and otolith signals indeed interact in the brain in order to separate head tilt
and linear acceleration [2, 6–12, 14–16]. The semicircular canals, however, are not the only
source of rotation information. Visual flow can also provide head rotation information since
full-field visual motion results from the sum of eye and head movements in space. Indeed,
visual information is believed to contribute to the computation of head tilt in space as vertical-
plane rotation of the visual field can create a strong illusory perception of tilt [17, 18] while
simultaneously inducing horizontal eye motions [19], both of which are indicative of a bias in
the estimation of gravity's direction.

Here we ask how the brain combines these two sources of rotation information to infer the
angular motion of the head for the purpose of verticality perception. A robust estimate of head
angular velocity can be achieved by combining visual and vestibular signals [20]. Because of
the statistical independence of the two sensory cues, the brain can increase the certainty of its
inferences by combining the two cues, each weighted by the inverse of their variance. Theory
predicts that by combining cues in this manner, the brain can achieve a statistically optimal
inference such that its variance is minimized [21, 22]. Near optimal combination of visual and
vestibular motion signals has been observed during translational self-motion perception [23],
(For review see: [24, 25]) and has been used successfully to model visual and vestibular interac-
tions during angular motion [26–28], suggesting the brain may indeed achieve an optimal
internal representation of the angular velocity of the head through a variance based weighting
scheme.

This weighting mechanism can be tested empirically by manipulating the variance of the
visual rotation signal by corrupting it with noise. According to the theory, increasing visual
noise will increase the visual cue's variance thereby reducing its weight when combined with
the semicircular canal signal. The head rotation velocity inferred should then shift away from
the velocity indicated by vision and towards the velocity indicated by the semicircular canals,
which, when the head is stationary, will be zero. Consequently, we predict that the addition of
visual noise will reduce illusory perception of tilt induced by visual-field rotation. Here we
show that when the noise is added to the rotating visual scene a reduction of apparent tilt does
indeed occur and a current model of visual-vestibular processing predicts both the main effect
of visual noise, but also the slow plateauing time course of apparent tilt development.

## Materials and methods

### Participants

10 healthy participants (4 male and 6 female between the ages of 21–39) with no known history
of neurological disease or injury participated in this study. Prior to participation, the experi-
mental protocol was explained and participant's informed written consent obtained. All proce-
dures in this study conformed to the declaration of Helsinki and were approved by the
National Research Ethics Service Committee.

### Procedure

Participants sat in the dark, 60cm in front of a 2.4m wide rear projection screen (The Wide-
screen Center Ltd, London, UK), with their head held in place with a stirrup shaped restraint
to limit motion. The visual stimulus was projected (Infocus DLP SP860, Portland, OR, USA)
on the screen as an annulus shaped field of dots (Fig 1A and 1B). The inner diameter of the
annulus was 48 cm (a visual angle of 44˚), the outer diameter of the annulus was 244 cm with a
visual angle of 128˚ and the annulus itself had a dot density of 1460 dots/m$^2$ with each coloured
dot being 1.2 cm (1.15˚) in diameter (**S1 File**). To indicate perceived vertical, participants con-
trolled the orientation of a linear sequence of 17 filled white circles (created in LabVIEW:

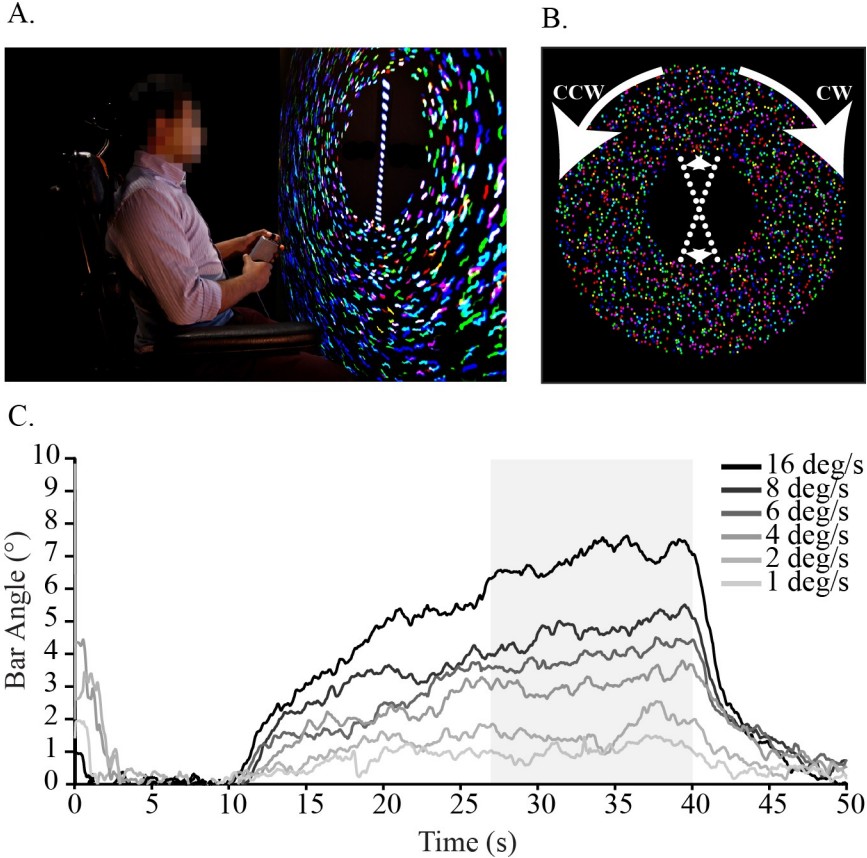

**Fig 1. Methods and experimental set up. A**. Participants sat in front of an annulus of colored dots **B**. The dots rotated clockwise (CW) or counter-clockwise (CCW) while participants controlled the angle of a dotted line in the center using a handheld potentiometer. **C**. Grand means (N = 10) time-course of bar bias for the zero noise conditions at each velocity. Time-period used to estimate mean bar bias during visual motion for each trial was from 27s to 40 seconds (shaded region).

National Instruments, Austin, TX, USA). Each circle had a diameter of 1.5 cm (visual angle of 1.4˚) and was 1.4 cm from the neighbouring circles. The linear sequence of filled circles was projected (Casio DLP data projector XJ533, Norderstedt, Germany) on to the center of the annulus and spanned the inner diameter of the annulus (Fig 1A and 1B).

The filled white circles for the bar were developed using custom Labview software and moved in rigid motion forming a segmented bar (Fig 1A and 1B). We used dots instead of a bar in order to limit the verticality cues that can arise due to pixilation along the edge of oblique lines. Participants were tasked with keeping the bar vertical for the duration of the trial using a hand-held potentiometer.

We added a small amount of low frequency noise (0–0.2Hz bandwidth, mean 0, standard deviation of 9.6˚, range +/-35.9˚) to the angular position of the bar during each trial to compel participants to continuously correct their perceived vertical. Trials began with the bar randomly oriented between ± 45˚, followed by a 10 s static period, when the dots were visible but not in motion. After the initial static period the dots rotated in either the clockwise or counterclockwise direction for 30s followed by a second 10s static period to finish the trial. A full trial lasted 50s and participants could control the angle of the bar for the full duration of the trial.

During trials, the annulus of dots could rotate at one of six velocities (1, 2, 4, 6, 8, 16˚/s) and at six noise levels (0, 25, 50, 60, 70, 80%), pseudo-randomly selected so that participants

completed four trials in each condition. However, because of limitations in our projection setup (dot duplication with jumps that were too large) we were unable to implement the full factorial design such that three conditions were omitted (16˚/s—80% noise, 16˚/s—70% noise and 8˚/s—80% noise). Within each testing session the angular velocity, direction and noise of the dots were varied randomly between trials for a total of 66 trials. To acquire sufficient trials (four of each condition in each direction) participants completed four separate testing sessions for a total of 264 trials. Visual stimuli were written using a custom Matlab program (Mathworks Inc., Natick, MA, USA) using the Psychophysics toolbox [29–31].

## Visual noise

In order to compare the effect of rotation velocity across different levels of noise, we added noise in a manner that preserves the average angular velocity of the visual scene (as described previously [32], S1 File). Briefly, to create the noise we treated each dot's change in position between frames as a vector ($\Omega$), the origin being its location at frame n and its tip being its location at frame n + 1. The magnitude of this vector was equivalent to the frame-by-frame dot jump necessary to maintain the desired angular velocity of the stimulus. We then added a random angular rotation to the vector around its origin. At 100% noise, the domain of the added rotation would be un-restricted (from 0 to 360˚) and would result in zero net motion over time, as jumps can occur in any direction with equal probability. As the noise is reduced, the domain of the angular rotation is reduced symmetrically around the direction of the desired net field motion (for example to between 160˚ and -160˚ if the direction of net field motion is 0˚) resulting in a residual motion vector ($\Omega_{net}$) in the direction of the desired stimulus motion. The vector $\Omega_{net}$ was then scaled with constant c so that $c\Omega_{net}$, the net angular velocity of the field of dots, equals $\Omega$, the desired angular velocity of the field. The magnitude of noise we could provide was limited by the asymptotic behaviour of the scale parameter used to scale the noise (it goes to $\infty$ at 100% noise). Therefore, the highest noise level tested was 80%.

## Data collection

Angular position of the bar was recorded at 30 Hz using a custom written Labview program and both the bar presentation/data collection Labview script and the Matlab stimulus presentation script were synchronized using custom written control software.

## Data reduction and analysis

In each condition the primary dependent measure was the angular deviation of the segmented bar from vertical (bar angle). In order to provide a simple single measure of bias we first zeroed the individual trials to zero by subtracting the trial's average value between 5 and 10 seconds from each data point. We then averaged each subject's data over the period between 27 and 40s, which coincides with the maximal response across conditions and a period of response saturation in the low velocity conditions (Fig 1C). To determine whether participants were biased more in one direction of motion than the other we compared the clockwise and counter-clockwise motion directions using the absolute value of the average bar angle from 27 to 40s. Bar angle was calculated for each subject and for each experimental condition separately, and the effect of direction was compared using paired t-tests with a Bonferroni adjusted p-value of 0.0015. Since the average bias of the two motion directions was not statistically different (All p > 0.01, only two of 33 comparisons were less than p = 0.05), we inverted the rebased time-series data from the clockwise trials and then averaged all the single trial time-series data together to get a single average time series for each condition in each participant. In total, eight trials were averaged (four inverted clockwise and four counter-clockwise) per

condition within each subject. In addition, we also fit the 16 ˚/s no noise condition in Matlab to provide a simple estimate of the exponential shape of subject's response to visual motion using an exponential of the form:

$$Y = a(1 - e^{-bx}) + c$$

where a, b and c are constants which were fit, x is the time variable and e is the exponential. The time constant was defined as the time taken for the exponential to reach 63% of its height.

## Statistical model

To determine whether changes in the angular velocity of, or noise in, the visual stimulus influenced the total bias accrued in perceived vertical, as indicated by the bar angle, we fit the average bar angle from 27 to 40s for all conditions using a linear mixed effects model in the R programming language [33] using the lme4 software package [34]. We treated the influence of each parameter (velocity, noise and their interaction) as a fixed effect and permitted the model's intercept to vary between subjects [35]. Improvements in the fit of the model with the addition of terms were compared using the likelihood-ratio test. Since we were only interested in whether these factors modulated perceived vertical we did not decompose the effect of these parameters further using pairwise comparison.

## Perceived velocity

To determine whether increases in visual noise are accompanied by a reduction in the perceived velocity of the visual stimulus we conducted an additional control experiment in eight participants (6 female, 2 male, 30 ± 6.5 yrs, with only one subject participating in both parts of the experiment). In this control experiment, participants completed a two alternative forced choice task where they were asked: is the second stimulus faster or slower than the first? One of the stimuli in the forced choice task was always a 6˚/s, zero-noise reference stimulus. The second stimulus served as a comparison and was pseudo-randomly selected from one of seven velocities (3, 4, 5, 6, 7, 8, 9 ˚/s) and one of four noise levels (0, 20, 40, 60%) so that each combination was drawn a total of 16 times (448 trials total). The order of presentation of the reference and comparison stimulus switched randomly from trial to trial. If participants perceived a noisy stimulus to be slower than a coherent stimulus, in line with the decreased effectiveness of noisy stimuli in biasing vertical, then the stimulus velocity perceived as equivalent to the 6˚/s coherent stimulus should increase as noise is added. To identify the point of subjective equality, the 7 velocities by 4 noise levels grid was interpolated to identify the contour line for the 50% decision threshold. Interpolation was performed because of the low probability that the 50% decision threshold would align with one of the conditions tested. If a single subject responded correctly 50% of the time at more than one velocity level for a given level of noise, as occurred in two participants, we averaged these velocities to create a single data point for that subject. To determine whether noise influenced the point of subjective equality, we fit a linear model to the perceptual data in R using the lme4 package [34] assuming a fixed effect for noise as well as a random slope for noise in each subject. Significance of the fixed effect for noise was assessed using the Wald test with the degrees of freedom estimated using Satterthwaite's method.

## Mechanistic model

To understand better potential mechanisms contributing to the bias in perceived vertical we used previously published model of visuo-vestibular processing to predict the expected bias in

vertical resulting from visual motion [36–38] (Fig 2). Briefly, we assumed that since participants were seated, the only stimuli present were the visual motion, in roll, and a vestibular encoding of gravity. The motion encoded by the brain was assumed to arise from slip of the visual scene relative to the retina [39]. This slip acts as a stimulus to drive motion of the eye to the perceived velocity of the head inferred by motion in the visual scene, with the objective of stabilizing the scene on the retina.

$$rSL(t) = Vis(t) - \Omega(t) \qquad \qquad \text{Eq 1}$$

Where $rSL(t)$ is the retinal slip, $Vis(t)$ is the three dimensional angular velocity of the scene and $\Omega(t)$ is the inferred angular velocity of the head. To estimate the angular velocity of the scene, which could imply angular motion of the head, retinal slip information is combined with angular motion cues from the vestibular system and feedback from gravio-inertial pathways via a leaky integration process [36]. This process constitutes the 'velocity storage' mechanism (VS) [36, 40]

$$\frac{dVS(t)}{dt} = k_o rSL(t) + k_v V(t) + -\frac{1}{T_{VS}} * VS(t) + k_f * (GIA(t) \; x \; G(t)) \qquad \text{Eq 2}$$

where $k_o$ is the retinal slip gain, $k_v$ is the vestibular gain, $V(t)$ is the vestibular signal, $T_{vs}$ is the leak time constant, $k_f$ is the rotation feedback gain and $GIA(t) \times G(t)$ is the rotation feedback (the cross product of the gravitoinertial acceleration {$GIA(t)$} and the central estimate of gravity {$G(t)$}). The output of the velocity storage is then summed with the visual and vestibular input to estimate inferred velocity of the head ($\Omega(t)$.)

$$\Omega(t) = G_o rSL(t) + G_v V(t) + VS(t) \qquad \qquad \text{Eq 3}$$

Where $G_o$ is the visual gain and $G_v$ is the vestibular gain. The inferred velocity of the head is then integrated to estimate the change in position of the head (and thereby quantify change in the inferred orientation of gravity) through Eq 4

$$\frac{dG(t)}{dt} = G(t) \; x \; \Omega(t) - \frac{1}{T_s} * (G(t) - GIA(t)) \qquad \text{Eq 4}$$

where $T_S$ is the somatogravic time constant. We assumed that participants aligned the bar they were controlling to indicate vertical with the inferred orientation of gravity $G(t)$. We fit the model to the data by minimizing the sum of squared error between the bias in gravity, produced by the model, and the across-participant average bar angle for all conditions. We limited model fitting to the visual ($K_o$, $G_o$) and vestibular ($K_v$, $G_v$) gain parameters and time constant $T_s$. The velocity storage time constant ($T_{vs}$) was set to 15s, similar to previous implementations of this model [35] Note: In a Bayesian framework, both the somatogravic and the velocity storage time constants can be conceptualized as prior distributions encapsulating the natural statistics of an individual's head movement. As such, the somatogravic time constant has been represented as a Gaussian prior centered at zero linear acceleration, and the velocity storage time constant has been represented as a Gaussian prior centered at zero angular velocity [35]. Since we were interested in evaluating the influence of noise on perceived vertical, once we fit the mechanistic model, we multiplied the visual and vestibular gains by 1—percent noise (1, 0.75, 0.5, 0.4, 0.3, 0.2) to determine whether the model would fit the mean empirical data for the noisy conditions. Model fit to the population mean for each condition was assessed using the coefficient of determination.

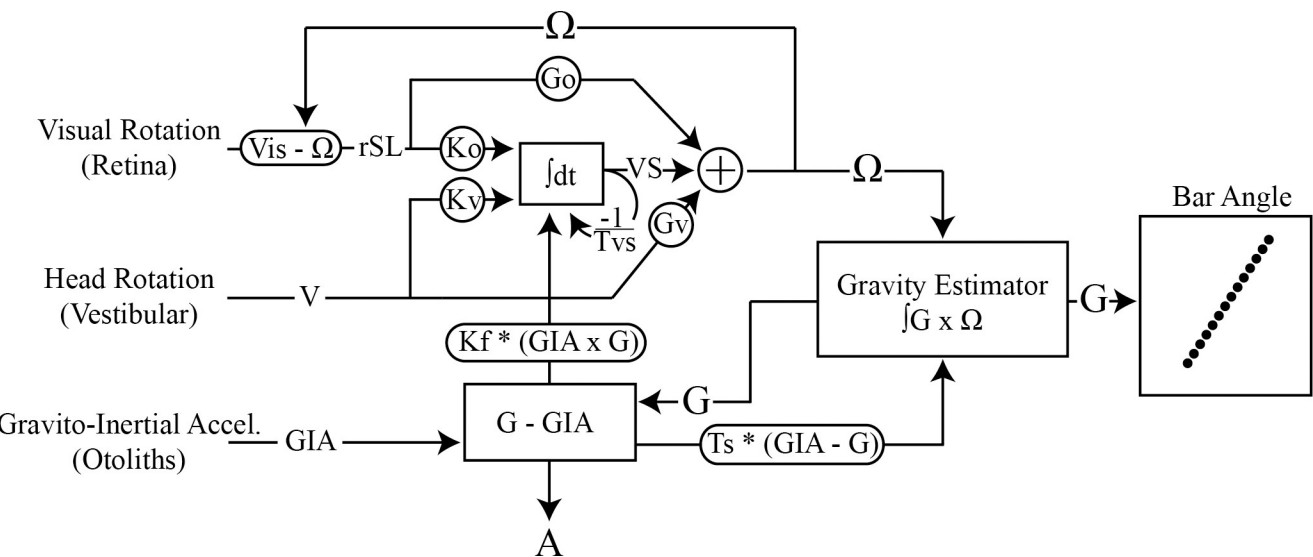

**Fig 2. Schematic of the mechanistic model describing the transformation of visual and vestibular motion to a change in bar angle.** Visual motion (Vis) is encoded as retinal slip (rSL), the difference between the internal estimate of the head's velocity and scenes velocity. The retinal slip information is multiplied by gain Ko (0.11*{1-percent noise}) and integrated overtime with vestibular signals (V) multiplied by gain Kv (0.2). The integration process is leaky with time constant Tvs (15) and is influenced by rotation feedback derived from the cross product of the Gravitoinertial acceleration signal (GIA) and the inferred orientation of gravity (G) multiplied by gain Kf (0.0). This integration process has been broadly described as velocity storage process. The output of the velocity storage is then summed with the rSL, multiplied by gain Go (0.16 *{1-percent noise}), and V, multiplied by gain Gv (0.43), to infer the angular velocity of the head (Ω). The cross-product of the inferred angular velocity of the head and the inferred gravity vector is then integrated to estimate the gravitational vector. This cross-product ensures that only rotations orthogonal to gravity are integrated. The difference between GIA and G is then used to estimate linear acceleration of the head (A) and acts as a negative feedback loop, that models the somatogravic effect, acting to pull G back into alignment with GIA (with gain factor Ts {0.74 }). For a more detailed description of this model see Laurens and Angelaki 2011 and Laurens et al., 2013a.

## Results

In general, perception of vertical became increasingly biased as the angular velocity of the visual scene increased and as the noise present in the visual stimulus decreased (Table 1). The inclusion of fixed effects for noise, velocity and their interaction improved the fit of the model over an intercept only model (Table 2). There was, however, high between-subject variance in this sample and for three participants there was very little modulating influence of noise or velocity. Participants whose perceived vertical was influenced more strongly by the stimulus also were influenced more by changes in velocity and by the addition of noise. In these participants, increases in visual motion velocity were accompanied by an increase in the bias of perceived vertical (Figs 3 and 4). Visual motion exerted its greatest influence on perceived vertical when stimulus noise was zero and stimulus velocity was highest. The opposite effect was observed with a progressive increase in visual noise, which tended to reduce the biasing effect of visual motion when the velocity of the visual scene was held constant (Figs 3 and 4). Noise had the largest effect when velocity was highest, and the smallest influence when velocity was lowest.

**Table 1. Grand mean bar angle for each condition and their standard deviations (n = 10) derived from the shaded region in Fig 1C.**

| | | | Noise | | | |
|---|---|---|---|---|---|---|
| Velocity | 80% | 70% | 60% | 50% | 25% | 0% |
| 1°/s | 0.4 ± 1.1 | 0.9 ± 1.1 | 1.2 ± 0.8 | 1.3 ± 1.3 | 1.3 ± 1.2 | 1.1 ± 1.1 |
| 2°/s | 0.8 ± 1.2 | 1.3 ± 2.0 | 1.9 ± 1.8 | 2.0 ± 1.2 | 2.0 ± 2.0 | 1.6 ± 2.5 |
| 4°/s | 0.9 ± 1.6 | 1.8 ± 1.8 | 2.2 ± 2.1 | 2.8 ± 2.7 | 2.8 ± 2.9 | 3.1 ± 2.5 |
| 6°/s | 1.2 ± 1.8 | 2.2 ± 2.3 | 2.9 ± 2.8 | 2.8 ± 3.0 | 4.0 ± 3.6 | 3.9 ± 4.0 |
| 8°/s | - | 2.3 ± 3.0 | 3.4 ± 2.9 | 3.7 ± 3.1 | 3.7 ± 4.1 | 4.7 ± 3.7 |
| 16°/s | - | - | 3.4 ± 5.1 | 5.0 ± 4.7 | 5.9 ± 5.8 | 7.0 ± 6.9 |

**Table 2. Likelihood ratio test results for the linear mixed effects model.**

| Likelihood Ratio Test | Comparison | Difference | Chisq | Df | P-value | Signif |
|---|---|---|---|---|---|---|
| Noise | Intercept Only | Fixed | 131.9 | 1 | $1.6e^{-30}$ | *** |
| Velocity | Intercept Only | Fixed | 43.4 | 1 | $4.4e^{-11}$ | *** |
| Velocity + Noise | Intercept Only | Fixed | 167.9 | 2 | $3.5e^{-37}$ | *** |
| Velocity : Noise | Intercept Only | Interaction Only | 7.3 | 1 | $6.8e^{-3}$ | ** |
| Velocity*Noise | Intercept Only | Interaction + Fixed | 178.8 | 3 | $1.6e^{-38}$ | *** |
| Velocity + Noise | Velocity*Noise | Interaction Added | 11.0 | 1 | $9.4e^{-4}$ | *** |
| Velocity : Noise | Velocity*Noise | Fixed Added | 171.5 | 2 | $5.7e^{-38}$ | *** |

** $p < 0.01$

*** $p < 0.001$

## Visual psychophysics

Since noise reduces the bias in perceived vertical, we investigated whether the addition of noise also altered participant's perception of the velocity of the stimulus. If participant's motion percept were the product of multisensory integration, we might expect that as noise is added to the stimulus the inferred motion velocity, a multisensory percept, would decrease, due to a decrease in visual weight. However, this was not the case. Instead the point of subjective equality exhibited a non-significant decreasing trend with the addition of noise to the visual scene (Fig 5) ($\beta$ = -1.6, $t_{(10)}$ = -1.96, $p$ = 0.078) suggesting, if anything, the addition of noise to the stimulus resulted in participants perceiving the stimulus velocity as faster rather than slower than it actually was.

## Visuo-vestibular model

To determine if the biasing effect of visual field motion on perceived vertical could be explained by known sensory cue combination mechanisms, we modelled the behavioural response to the stimulus using a prominent visual-vestibular processing model (Fig 2). To predict the result of changing visual stimulus velocity we input the six stimulus velocities into the model (1, 2, 4, 6, 8, 16˚/s). We also estimated the influence of adding noise to the visual stimulus by multiplying the visual gains ($K_o$, $G_o$) in the model by one minus the percentage of added noise (1, 0.75, 0.5, 0.4, 0.3, 0.2). The mechanistic model predicted the asymptotic behavior of participant's responses for velocity (Fig 6A) and each level of noise (Fig 6B). However, the fit was generally better for higher-velocity low-noise conditions than it was for low-velocity high-noise conditions. The model also tended to slightly underestimate the average bias across most velocities and exhibited asymptotic behavior earlier in time than participant's data. Overall, our results suggest the asymptotic behavior observed during subjective vertical could be explained by an equilibrium reached between feedback acting on the tilt estimator, rotation feedback in the velocity storage and the biasing effect of the visual motion stimulus.

## Discussion

The purpose of this study was to investigate whether biases in perceived vertical caused by visual field motion are reduced when noise in the visual field is introduced. We found that both the addition of noise and a reduction in visual motion velocity reduce the impact of visual field motion on perceived vertical. In addition, we found participants' perception of the velocity of the visual stimulus was not significantly affected by the addition of noise, suggesting that motion information is processed differently for motion perception and tilt estimation. Lastly,

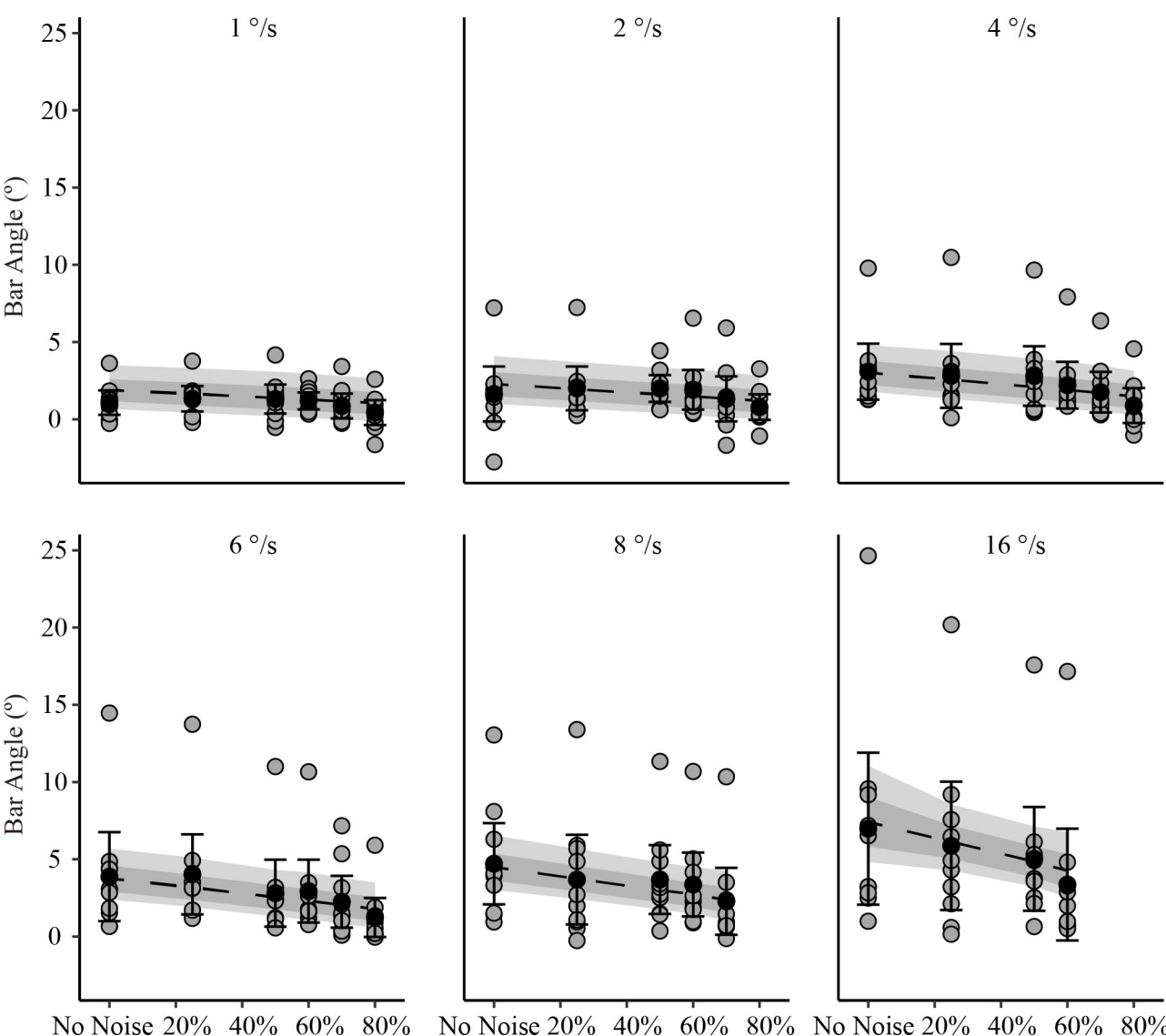

**Fig 3. Linear multilevel mixed effects model with sample mean and individual data displaying the effect of adding noise on perceived vertical.**
Noise decreased the influence on visual motion on perceived vertical but the size of the effect depends on the velocity. Each subject's mean bar angle for the last 13s of visual motion in each trial are shown as grey circles. Sample means are black circles (n = 10). The error bars are standard deviations. The segmented line is the bootstrapped linear multilevel mixed effects model mean and the shaded regions are the 68% and 95% confidence intervals for the mean.

participants' behaviour over time could be explained by mechanisms of multisensory cue combination which incorporate negative feedback acting on the brains' estimate of velocity and tilt relative to gravity, which limits the extent of the bias in perceived vertical during prolonged periods of visual motion [37].

## The influence of visual noise on perceived vertical

As predicted by the principles of multisensory integration, the addition of noise to the visual stimulus reduced the influence of visual motion on perceived vertical, which could be modelled by a decrease in the weight of visual information during processing. Since this visual motion information is used to derive a central estimate of rotational velocity of the head which

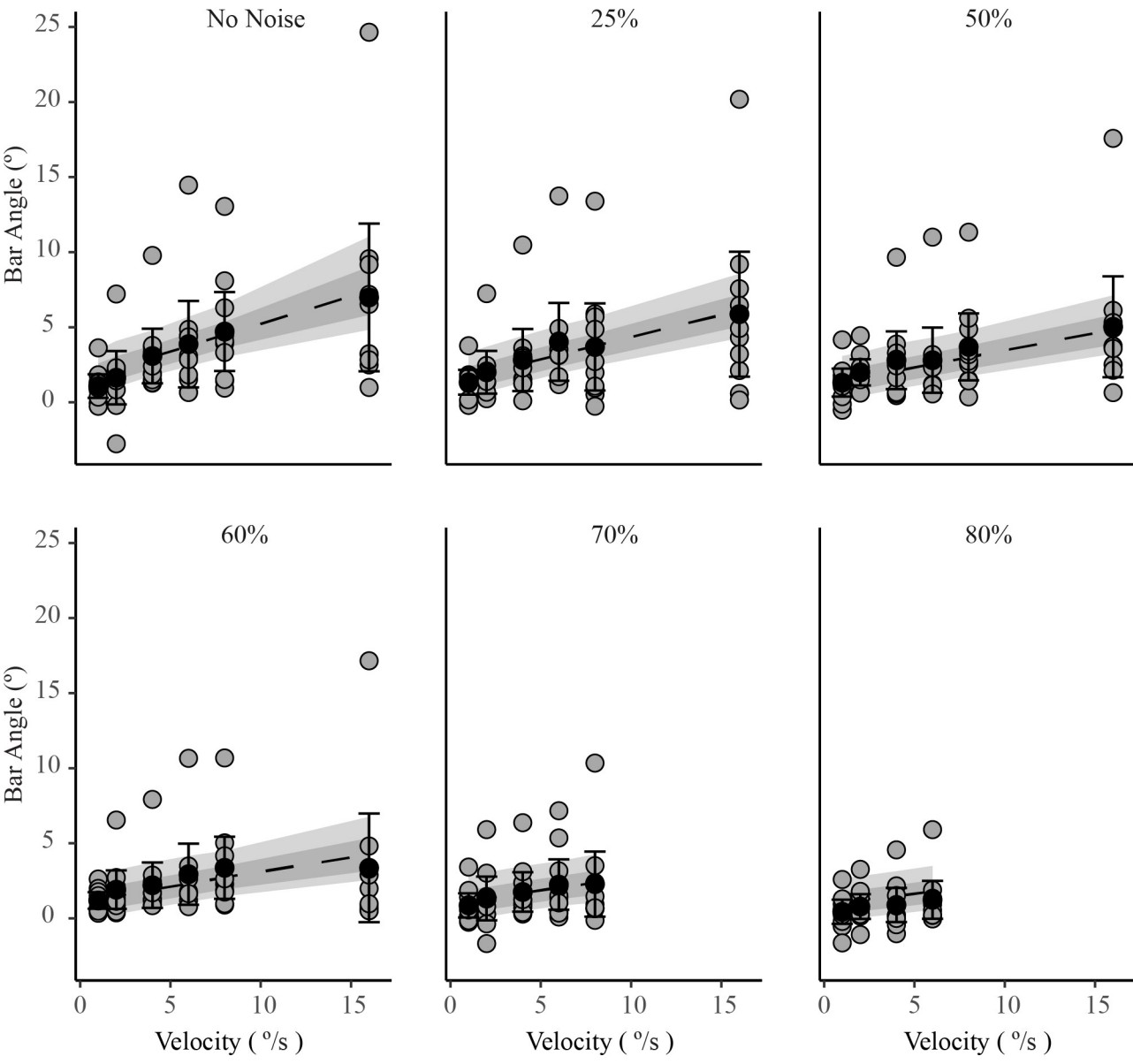

**Fig 4. Linear multilevel mixed effects model with sample mean and individual data displaying the effect of increasing velocity on perceived vertical.** Higher velocities biased vertical more than lower velocities and the size of the effect depends on the noise level. Each subject's mean bar angle for the last 13s of visual motion in each trial are shown as grey circles. Sample means are black circles (n = 10). The error bars are standard deviations. The segmented line is the bootstrapped linear multilevel mixed effects model mean and the shaded regions are the 68% and 95% confidence intervals for the mean.

could inform our conscious experience it was unclear whether the addition of noise would also result in an underestimation of the perceived velocity of the visual stimulus. However, we found no evidence to support this possibility when we formally measured the effect of visual noise on the perception of visual rotation velocity. If anything, the effect occurred in the opposite direction as there was a trend towards perceiving the stimulus as faster when noise was added. Such a trend could be explained by local dot motion dynamics. Specifically, while the mean field velocity of the stimulus remained constant, the dot's jump distance from frame to frame increased with the addition of noise resulting in greater apparent motion for any specific

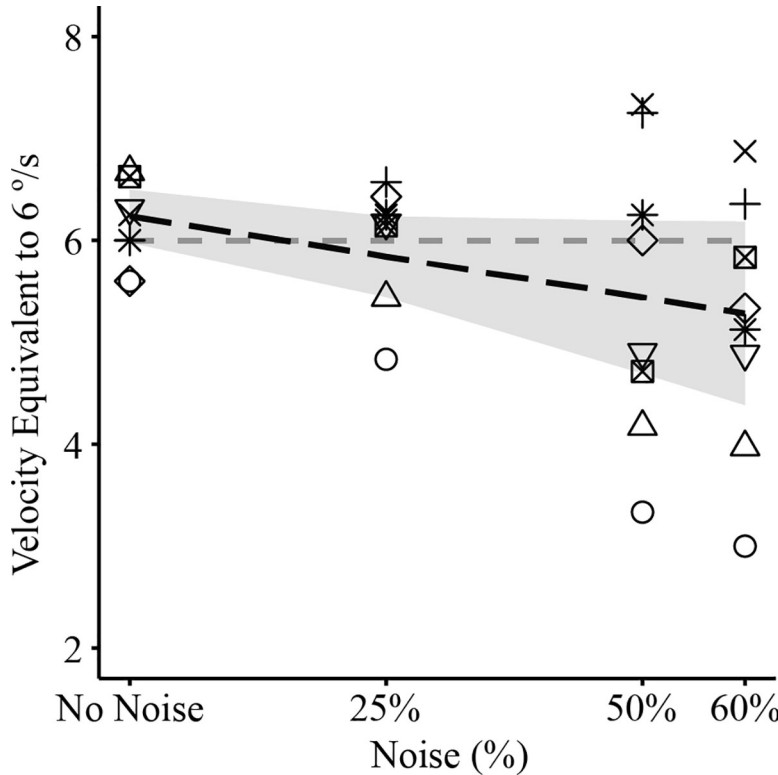

**Fig 5. Point of subjective equality between the reference 6 °/s stimulus (horizontal grey dotted line) and a comparison stimulus with different levels of noise (n = 8).** With the addition of noise, the point of subjective equality becomes much more variable across participants and exhibits a non-significant ($\beta$ = -1.6, $t_{(10)}$ = -1.96, p = 0.078) decreasing trend, opposite of what is expected if participants perceived the stimulus as slower with added noise. Each subject has its own symbol, the dotted line indicates the mean, and large filled dark circles are the means at each noise level. The shaded region is the 95% confidence interval for the mean.

dot between consecutive frames. This observation suggests that the local kinematics of a dot's motion might influence global motion perception and that the noise isn't completely 'integrated out' when dot motion information is pooled over space or time. One limitation of our results in light of this interpretation is that participants may have attended to local dot motion rather than the global field motion. The contrasting effects of the addition of visual noise on motion perception and gravity estimation suggests that their respective visual processing mechanisms may differ or that the visual motion data may be decoded differently for these two mechanisms.

## Asymptotic behaviour of perceived vertical over time

Prolonged presentation of angular visual field motion resulted in a bias in perceived vertical that reached an asymptotic value with a time constant of approximately 11.4 s in the 16 °/s–0% noise condition. Moreover, the saturation in bias of perceived vertical over time can be explained by feedback loops acting on the tilt estimation process and the velocity estimate [38]. As the perceived orientation of gravity separates from the otolith's gravito-inertial force signal, two sources of feedback act to limit further bias in perceived vertical [10, 11, 35–37, 41]. The first acts directly on the tilt estimation process while the other acts indirectly by adjusting the inferred velocity of the head.

The direct feedback loop's purpose is to eliminate drift and recalibrate our tilt estimate. Such a mechanism is necessary because the summation of the inferred angular velocity of the

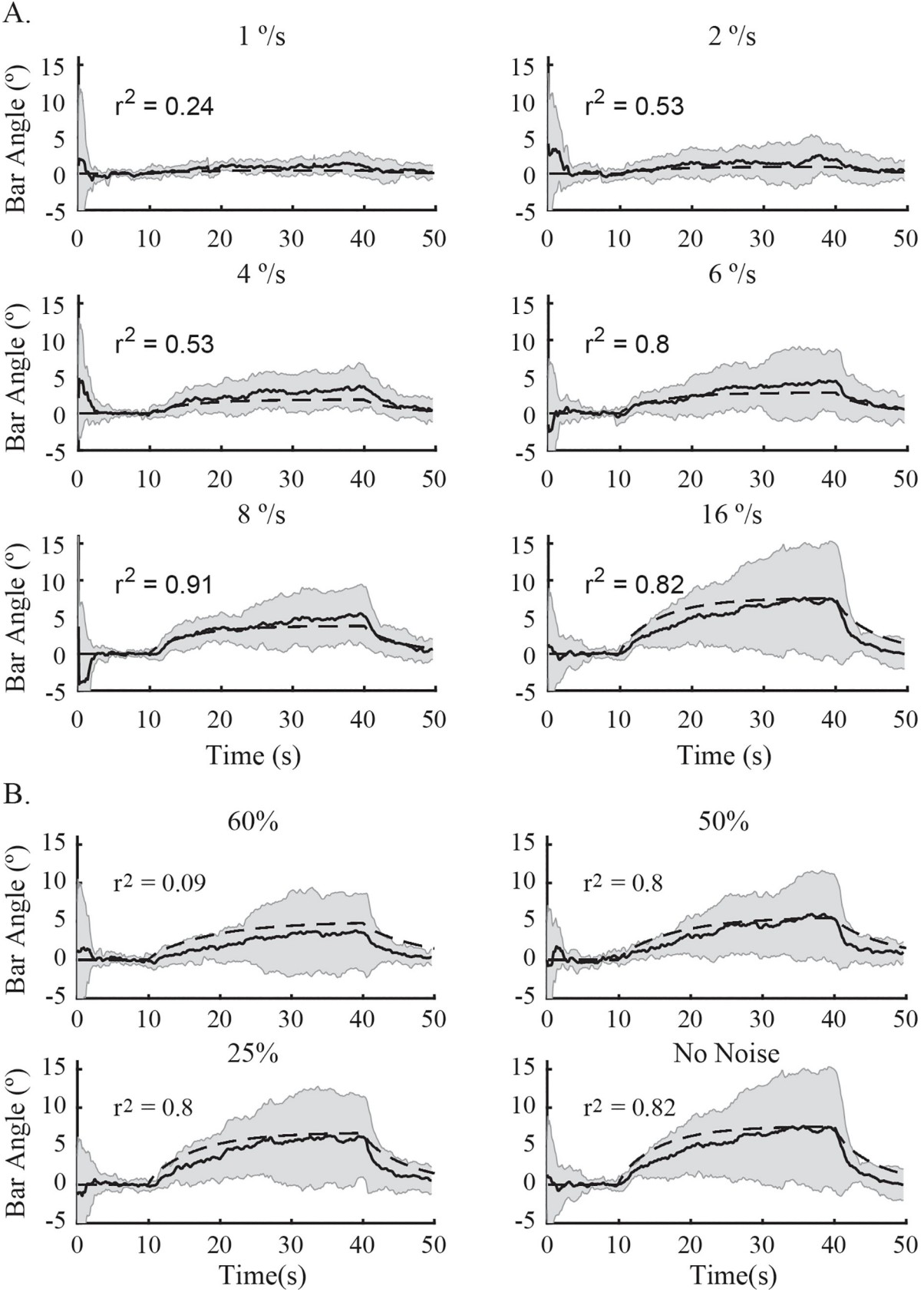

**Fig 6. Comparison of data with the model (n = 10) for different velocities with noise held constant (zero noise) and different levels of noise with a velocity held constant at (16°/s). A.** Solid black line is the mean bar angle across participants whereas the shaded area is the standard deviation of participant's mean response. The segmented line displays the model's behaviour. At low velocities the model slightly underestimates the bar angle and it appears to have a shorter time constant than participant's responses. Model fit to the grand mean is displayed as the $r^2$ value on each plot. **B.** We simulated gain changes with visual noise by multiplying the visual pathway gains (Ko, Go) by one minus the percentage of noise (1, 0.75, 0.5, 0.4, 0.3, 0.2). Model correspondence to the grand mean is displayed as the $r^2$ value on each plot. Shaded area is the standard deviation of participant's mean response.

head over time, which is noisy, can lead to drift in the estimate of the orientation of gravity. To counter this drift, the direct (somatogravic) feedback loop acts slowly to pull the estimated orientation of gravity back into alignment with the otolith's signal. Because the direct feedback loop operates on a slow time scale, its influence on transient events, such as most translational accelerations, is thought to be limited [35]. The indirect feedback loop, which is often incorporated in visual-vestibular processing, acts to adjust the inferred angular velocity of the head, indirectly leading to a reduction in the integration rate of the tilt estimator. Much like the direct feedback loop, the indirect feedback loop's influence increases as the separation between the estimated orientation of gravity and the gravito-inertial acceleration signal encoded by the otolith signals increases. During the presentation of angular visual motion on its own, the lack of accompanying head tilt results in a divergence of the estimated orientation of gravity and the otolith signal. The indirect feedback mechanism acts to reduce the angular velocity inferred by the brain thereby reducing the rate of divergence between the estimated orientation of gravity and the otolith signal, resulting in a decrease in the rate of bias of the perceived orientation of gravity.

This visual-vestibular processing model can also be conceptualized as a recursive Bayesian model with priors for zero angular velocity and linear acceleration [9, 26, 35, 42]. Such Bayesian priors reflect an adaptation of the brain to the natural statistics of our motion. Namely, since prolonged non-zero linear accelerations and angular velocities are improbable, over time the brain's estimate of angular velocity and linear acceleration will decay towards zero. During prolonged linear acceleration this latter 'acceleration prior' will cause the motion perceived to transition slowly from a perception of linear acceleration to a perception of tilt, due to linear acceleration's yoking to the estimate of gravity (tilt) and tilt being a statistically more probable event than prolonged linear acceleration [9, 26, 42].

## Limitations

The amount of bias in perceived vertical caused by visual motion is highly dependent upon the age of the sample. As we age, we become more susceptible to the influence of visual motion on perceived vertical, and therefore modulating factors like changes in velocity and added noise have greater influence [32, 43]. Here we examined visual motion induced bias in vertical in young adults, which have small average responses. Changes in bias due to our independent variables (velocity and noise) were therefore very small in some participants, particularly so at lower velocities. We also added low frequency noise to the bar indicating participant's perceived vertical in order to compel them to continuously adjust the orientation of the bar. This additional noise increased the variance of participant's single trial responses, to a degree depending on the studiousness of the participant, further reducing the fit of the mechanistic model. Together these factors may have reduced resolution of the different conditions and thus impeded comparison to the mechanistic model.

To examine the effect of noise on perceived vertical over time we used a model formulation similar to that used recently by MacNeilage and Glasauer [38]. Alternatively, a Bayesian formulation could also be used to describe this behaviour and indeed formulations of both types

of models have been proposed by Laurens and colleagues [26, 35, 36, 42] and both ultimately produce similar outcomes under a range of conditions. Here we chose to reduce only visual weighting to account for the effect of noise. However, in a Bayesian formulation the weights of the contributing inputs are normalized by the total variance, which effectively couples the weights. Consequently, such coupling increases reliance on the unaffected sensory modalities above that expected if the weights were uncoupled.

## Conclusion

Here we have demonstrated that current models of visual- vestibular processing may explain biases in perceived vertical induced by visual field rotation with varying levels of noise. Specifically, the slow change in verticality bias towards an asymptotic value during prolonged exposure could be explained by an equilibrium being reached between two processes; integration of the head rotation estimate driving the gravity estimate away from the otoliths' signal and the pull of feedback mechanisms dragging it back into alignment. Overall, these findings further our knowledge of the mechanisms underlying human verticality perception and provide a means of generating and testing hypotheses for its disruption in neurological disease.

## Supporting information

**S1 File. Example of the visual stimulus for a counter-clockwise, 60% noise, 16 ˚/s condition.** The 6 second video is in real time.
(MP4)

## Acknowledgments

We thank Jean-Sébastien Blouin for discussion on the mechanistic model.

## Author Contributions

**Conceptualization:** Christopher J. Dakin, Brian L. Day.

**Data curation:** Christopher J. Dakin.

**Formal analysis:** Christopher J. Dakin, Prateek Kumar, Patrick A. Forbes.

**Funding acquisition:** Christopher J. Dakin.

**Investigation:** Christopher J. Dakin, Prateek Kumar, Amy Peters, Brian L. Day.

**Methodology:** Christopher J. Dakin, Patrick A. Forbes, Amy Peters, Brian L. Day.

**Project administration:** Christopher J. Dakin, Amy Peters, Brian L. Day.

**Resources:** Brian L. Day.

**Software:** Christopher J. Dakin, Patrick A. Forbes.

**Supervision:** Christopher J. Dakin, Brian L. Day.

**Visualization:** Christopher J. Dakin, Prateek Kumar, Brian L. Day.

**Writing – original draft:** Christopher J. Dakin, Prateek Kumar, Brian L. Day.

**Writing – review & editing:** Christopher J. Dakin, Prateek Kumar, Patrick A. Forbes, Amy Peters, Brian L. Day.

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
