## [Decision Letter · Decision Letter 0]

16 Sep 2019

PONE-D-19-20795

Variance based weighting of multisensory head rotation signals for verticality perception

PLOS ONE

Dear Dr. Dakin,

Thank you for submitting your manuscript to PLOS ONE. After careful consideration, we feel that it has merit but does not fully meet PLOS ONE’s publication criteria as it currently stands. Therefore, we invite you to submit a revised version of the manuscript that addresses the points raised during the review process.

ACADEMIC EDITOR: Both reviewers found that the study has merit. Please follow the reviewers' suggestion just to improve the clarity of your manuscript for the readers' sake.

We would appreciate receiving your revised manuscript by Oct 31 2019 11:59PM. To enhance the reproducibility of your results, we recommend that if applicable you deposit your laboratory protocols in protocols.io, where a protocol can be assigned its own identifier (DOI) such that it can be cited independently in the future. For instructions see: http://journals.plos.org/plosone/s/submission-guidelines#loc-laboratory-protocols

We look forward to receiving your revised manuscript.

Kind regards,

Kei Masani

Academic Editor

PLOS ONE

Journal Requirements:

We thank Jean-Sébastien Blouin for discussion on the mechanistic model. C.J.D. was supported by a Canadian Institutes for Health Research Postdoctoral Fellowship. P.A.F. received funding from the Netherlands Organization for Scientific Research (NWO #016.Veni.188.049).

No: The funders had no role in study design, data collection and analysis, decision to publish, or preparation of the manuscript.

Reviewers' comments:

Reviewer's Responses to Questions

**Comments to the Author**

1. Is the manuscript technically sound, and do the data support the conclusions?

Reviewer #1: Partly

Reviewer #2: Partly

2. Has the statistical analysis been performed appropriately and rigorously? 

Reviewer #1: Yes

Reviewer #2: I Don't Know

3. Have the authors made all data underlying the findings in their manuscript fully available?

Reviewer #1: Yes

Reviewer #2: Yes

4. Is the manuscript presented in an intelligible fashion and written in standard English?

Reviewer #1: Yes

Reviewer #2: Yes

5. Review Comments to the Author

Reviewer #1: In this manuscript the authors study a well known illusion. Facing a rotating cloud of dots the subjects report an illusory tilt of the subjective vertical. The authors interpret the results in terms of multi-sensory integration and introduce noise in the visual stimuli to achieve their goal. I think the data are slightly over-interpreted and nuance would benefit the paper (see below). However, the paper is well written, the data are appropriately analyzed and the results consistent.

The authors argue that the illusion is the consequence of multi-sensory integration. However, it appears to be a simple visual illusion (and shows how the visual input "dominates" the other sens). While the proprioceptive, touch/pressure, vestibular cues are telling you that you are upright, the perceptual process relies on the visual input creating an illusory tilt. Whether the visual input becomes less reliable (noisier) the system starts using other sources of information such as the vestibular cues and the illusion decreases. Neural response has been observed in this sens in MSTd during translation (by Angelaki and colleagues). MSTd neurons that respond to visual and vestibular stimuli will preferentially encode the most reliable stimulus.

This interpretation of the data should be mentioned in the discussion section.

The first sentence of the manuscript would also benefit from more accurate wording: the otoliths do not signals "the body orientation in the gravitational field". Otoliths afferent respond only to linear accelerations.

Beside this minor revisions the paper is easy and pleasant to read.

Reviewer #2: This manuscript describes an experiment investigating the effect of a roll rotation visual stimulus on the subjective visual vertical, and how it depends on the noise associated with the visual velocity estimate. The conclusion is that bias in perceived upright that is induced by the rotating visual stimulus increases with increasing velocity (which has been shown before) and decreases as noise is added to the direction information in the visual motion stimulus.

The study is well-designed and addresses an interesting and important question in the perception of spatial orientation. I have suggestions for improved clarity in some places and other modifications.

Line 48, cite also Glasauer, S. (1992). Interaction of semicircular canals and otoliths in the processing structure of the subjective zenith. Ann NY Acad Sci, 656, 847-849.

65, cite also Jürgens, R., & Becker, W. (2006). Perception of angular displacement without landmarks: evidence for Bayesian fusion of vestibular, optokinetic, podokinesthetic, and cognitive information. Experimental Brain Research, 174(3), 528-543.

110, “radially scaled noise levels” is explained later, but confusing when presented here with no other context. Please clarify somehow.

111, please be explicit about what the limitation were. If others would like to reproduce your paradigm, this information would be useful

160 Is there a heading missing here?

161 to 170: Since this is the analysis that is used to examine the significance of the effect of both noise and velocity, which are the central findings of this paper, more explanation would be helpful. Specifically, it would be helpful to have more clarity on what exactly this analysis is testing. In laymen’s terms, what is the logic underlying this analysis?

183 to 187, “the data were interpolated” please clarify. I do not understand what type of analysis would yield two points with 50% performance.

188: Is this the same a fitting a line and asking whether the slope is significantly different from zero?

228: Somewhere in here, note that the two time constant can be conceptualized as Bayesian priors for zero angular velocity and zero linear acceleration (Laurens & Angelaki, 2011).

232 to 235: This explanation of the data in terms of the fixed effect model and the influence of high between subject variance is opaque. Please clarify.

251: Suggests that people were simply judging the visual velocity of the stimulus in this task, not the self-motion velocity

259: I am not familiar with the term “procession model”

267: What happens if you attempt to fit the model to all the data rather than simply to the 16 deg/s data?

277: This is not surprising to me. If you had asked subjects to judge the velocity of self-motion rather than visual motion, you might have gotten different results.

299: “visual processing mechanisms differ” Alternatively, common visual motion processing, but different read-out of those signals, i.e. visual motion versus self-motion

303: “over time can be”

308 to 350: Seems a false dichotomy is presented here. These can all be conceptualized as Bayesian models (Laurens & Angelaki 2011)

6. PLOS authors have the option to publish the peer review history of their article (what does this mean?). If published, this will include your full peer review and any attached files.

Reviewer #1: Yes: Jerome Carriot

Reviewer #2: No

---

## [Author Response · Author response to Decision Letter 0]

13 Nov 2019

Reviewer #1: In this manuscript the authors study a well known illusion. Facing a rotating cloud of dots the subjects report an illusory tilt of the subjective vertical. The authors interpret the results in terms of multi-sensory integration and introduce noise in the visual stimuli to achieve their goal. I think the data are slightly over-interpreted and nuance would benefit the paper (see below). However, the paper is well written, the data are appropriately analyzed and the results consistent.

The authors argue that the illusion is the consequence of multi-sensory integration. However, it appears to be a simple visual illusion (and shows how the visual input "dominates" the other sens). While the proprioceptive, touch/pressure, vestibular cues are telling you that you are upright, the perceptual process relies on the visual input creating an illusory tilt. Whether the visual input becomes less reliable (noisier) the system starts using other sources of information such as the vestibular cues and the illusion decreases. Neural response has been observed in this sens in MSTd during translation (by Angelaki and colleagues). MSTd neurons that respond to visual and vestibular stimuli will preferentially encode the most reliable stimulus.

This interpretation of the data should be mentioned in the discussion section.

We appreciate the reviewer's comment and agree that references to the important work from Angelaki and colleagues can help provide a neural explanation to our current observations. Accordingly we have made changes on lines 344-347. In these efforts we have kept our explanation and interpretation of the data in line with those used for multisensory integration models by Angelaki et al.. Specifically, that a shift in the weighting of different sensory sources is commonly observed when the reliability of these sensory sources changes, whereby such a shift results in a preferential encoding of the more reliable stimulus. We further note that our method of varying the visual weights differs from standard cue integration models. To highlight this difference we have highlighted the following to the limitations section: "Here we chose to reduce only visual weighting to account for the effect of noise however, in such a Bayesian formulation the weights of the contributing inputs are normalized by the total variance, which effectively couples the weights. Consequentially, such coupling increases reliance on the unaffected sensory modalities above that expected if the weights were uncoupled." 

The first sentence of the manuscript would also benefit from more accurate wording: the otoliths do not signals "the body orientation in the gravitational field". Otoliths afferent respond only to linear accelerations.

We agree with the reviewer and have revised this sentence to read:" The brain uses multisensory information to estimate the direction of gravity, in part because sensory organs that respond to linear acceleration, e.g. otoliths, also signal body orientation in the gravitational field due to Einstein's equivalence principle [1]. "

Beside this minor revisions the paper is easy and pleasant to read.

Reviewer #2: This manuscript describes an experiment investigating the effect of a roll rotation visual stimulus on the subjective visual vertical, and how it depends on the noise associated with the visual velocity estimate. The conclusion is that bias in perceived upright that is induced by the rotating visual stimulus increases with increasing velocity (which has been shown before) and decreases as noise is added to the direction information in the visual motion stimulus.

The study is well-designed and addresses an interesting and important question in the perception of spatial orientation. I have suggestions for improved clarity in some places and other modifications.

Line 48, cite also Glasauer, S. (1992). Interaction of semicircular canals and otoliths in the processing structure of the subjective zenith. Ann NY Acad Sci, 656, 847-849.

We have added this citation to Line 44

65, cite also Jürgens, R., & Becker, W. (2006). Perception of angular displacement without landmarks: evidence for Bayesian fusion of vestibular, optokinetic, podokinesthetic, and cognitive information. Experimental Brain Research, 174(3), 528-543.

We have added this citation to Line 61

110, “radially scaled noise levels” is explained later, but confusing when presented here with no other context. Please clarify somehow.

To avoid being redundant over successive paragraphs and to improve clarity we removed mention of the radial scaling at line 110 because of its lack of context, and left the detailed description of the noise to lines 120 – 134. 

111, please be explicit about what the limitation were. If others would like to reproduce your paradigm, this information would be useful

We included further description of the limitation. Line 107 now reads: "However, because of limitations in our projection setup (dot duplication during large jumps at high velocities) we were unable to implement the full factorial design such that three conditions were omitted (16 °/s - 80% noise, 16 °/s - 70% noise and 8 °/s - 80% noise)."

160 Is there a heading missing here?

We added the heading "Statistical Model" to this section

161 to 170: Since this is the analysis that is used to examine the significance of the effect of both noise and velocity, which are the central findings of this paper, more explanation would be helpful. Specifically, it would be helpful to have more clarity on what exactly this analysis is testing. In laymen’s terms, what is the logic underlying this analysis?

To simplify the statistical analysis and reduce the likelihood of overfitting the statistical model we reduced the complexity of the statistical model. The mixed effects model now tests the hypothesis that increases in velocity and decreases in noise will increase the biasing effect of the stimulus with a unique intercept in each subject. We conservatively assumed the simplest relationship (linear) between change in velocity and noise. We revised the wording of this section to read “To determine whether changes in the angular velocity of, or noise in, the visual stimulus influenced the total bias accrued in perceived vertical, as indicated by the bar angle, we fit the average bar angle from 27 to 40s for all conditions using a linear mixed effects model in the R programming language [33] using the lme4 software package [34]. We treated the influence of each parameter (velocity, noise and their interaction) as a fixed effect and permitted the model's intercept to vary between subjects. Improvements in the fit of the model with the addition of terms were compared using the likelihood-ratio test. Outcome of the likelihood ratio tests are presented in Table 2. Since we were only interested in whether these factors modulated perceived vertical we did not decompose the effect of these parameters further using pairwise comparison." 

Below are the new Figures 3 & 4 with the simplified model fits:

183 to 187, “the data were interpolated” please clarify. I do not understand what type of analysis would yield two points with 50% performance.

The data formed a grid: 7 velocities by 4 noise levels, with each point in the grid being the percent correct after 16 trials. Because of the discrete nature of the grid it is unlikely that the 50% condition will lie exactly on one of the conditions provided therefor the velocity and noise level for the point of subjective equality was estimated through interpolation to get the 50% contour across all conditions. If, a single subject had two velocities at a specific noise level in which they exhibited the same 50% performance we assumed the midpoint between these two velocities to be the 50% point (interpolation).

To clarify this section in the manuscript we added we reworded it to read: " To identify the point of subjective equality, the 7 velocities by 4 noise levels grid was interpolated to identify the contour line for the 50% decision threshold. Interpolation was performed because of the low probability that the 50% decision threshold would align with one of the conditions tested. If a single subject responded correctly 50% of the time at more than one velocity level for a given level of noise, as occurred in two participants, we averaged these velocities to create a single data point for that subject."

188: Is this the same a fitting a line and asking whether the slope is significantly different from zero?

Essentially, but instead of modelling the group behavior directly, we modelled each subject independently, permitting each subject to have a different intercept and slope, from which we estimated the group behavior.

228: Somewhere in here, note that the two time constant can be conceptualized as Bayesian priors for zero angular velocity and zero linear acceleration (Laurens & Angelaki, 2011).

We have included this information into this section. It now reads: " The velocity storage time constant (Tvs) was set to 15s, similar to previous implementations of this model [35] (Note: In a Bayesian framework, both the somatogravic and the velocity storage time constants can be conceptualized as prior distributions encapsulating the natural statistics of an individual's head movement. As such, the somatogravic time constant has been represented as a Gaussian prior centered at zero linear acceleration, and the velocity storage time constant has been represented as a Gaussian prior also centred at zero angular velocity [35]. Since we were interested in evaluating the influence of noise on perceived vertical, once we fit the mechanistic model, we multiplied the visual and vestibular gains by 1 - percent noise (1, 0.75, 0.5, 0.4, 0.3, 0.2) to determine whether the model would fit the mean empirical data for the noisy conditions. Model fit to the population mean for each condition was assessed using the coefficient of determination (Figure 6)."

232 to 235: This explanation of the data in terms of the fixed effect model and the influence of high between subject variance is opaque. Please clarify.

We revised this section to read: “In general, perception of vertical became increasingly biased as the angular velocity of the visual scene increased and as the noise present in the visual stimulus decreased. The inclusion of fixed effects for noise, velocity and their interaction (Table 2) improved the fit of the model over an intercept only model. There was, however, high between-subject variance in this sample and for three participants there was very little modulating influence of noise or velocity. Participants whose perceived vertical was influenced more strongly by the stimulus also were influenced more by changes in velocity and by the addition of noise. In these participants, increases in visual motion velocity were accompanied by an increase in the bias of perceived vertical (Figure 3 & 4). Visual motion exerted its greatest influence on perceived vertical when stimulus noise was zero and stimulus velocity was highest. The opposite effect was observed with a progressive increase in visual noise, which tended to reduce the biasing effect of visual motion when the velocity of the visual scene was held constant (Figure 3 & 4). Noise had the largest effect when velocity was highest, and the smallest influence when velocity was lowest." 

251: Suggests that people were simply judging the visual velocity of the stimulus in this task, not the self-motion velocity

We have added a sentence to the discussion to acknowledge this possibility: " Such a trend could be explained by local dot motion dynamics. Specifically, while the mean field velocity of the stimulus remained constant, the dot's jump distance from frame to frame increased with the addition of noise resulting in greater apparent motion for any specific dot between consecutive frames. This observation suggests that the local kinematics of a dot's motion might influence global motion perception and that the noise isn't completely 'integrated out' when dot motion information is pooled over space or time. One limitation of our results in light of this interpretation is that participants may have attended to local dot motion rather than the global field motion."

259: I am not familiar with the term “procession model”

We have changed this to: "processing model"

267: What happens if you attempt to fit the model to all the data rather than simply to the 16 deg/s data?

We thank the reviewer for this suggestion and have replaced figure 6 with a model fit to the full data set (Figure to the right). By fitting the model to the full data set, the correspondence between the model fit and the mean of the conditions presented in Figure 6 is reduced (an overall r2 of 0.78 versus 0.84). Presumably, this is because this model fits the conditions with low signal to noise ratio's better at the expense of the conditions with high signal to noise ratio. 

277: This is not surprising to me. If you had asked subjects to judge the velocity of self-motion rather than visual motion, you might have gotten different results.

We revised the wording of this sentence to read: "In addition, we found participants’ perception of the velocity of the visual stimulus was not significantly affected by the addition of noise, suggesting that motion information is processed differently for motion perception and tilt estimation."

299: “visual processing mechanisms differ” Alternatively, common visual motion processing, but different read-out of those signals, i.e. visual motion versus self-motion

We have changed this sentence to read: "The contrasting effects of the addition of visual noise on motion perception and gravity estimation suggests that their respective visual processing mechanisms may differ or that the visual motion data may be decoded differently for these two mechanisms."

303: “over time can be”

Fixed, thank you!

308 to 350: Seems a false dichotomy is presented here. These can all be conceptualized as Bayesian models (Laurens & Angelaki 2011)

We have reworded this a section to try to prevent generating the impression of a dichotomy. The section now reads: "The direct feedback loop's purpose is to eliminate drift and recalibrate our tilt estimate. Such a mechanism is necessary because the summation of the inferred angular velocity of the head over time, which is noisy, to can lead to drift in the estimate the orientation of gravity. To counter this drift, the direct (somatogravic) feedback loop acts slowly to pull the estimated orientation of gravity back into alignment with the otolith's signal. Because the direct feedback loop operates on a slow time scale, its influence on transient events, such as most translational accelerations, is thought to be limited [36]. However, during prolonged linear accelerations the direct feedback loop will cause the motion perceived to transition slowly from linear acceleration to tilt as a result of the acceleration estimates' yoking to the estimate of gravity [9, 26, 43]. The indirect feedback loop, which is often incorporated in visual-vestibular processing, acts to adjust the inferred angular velocity of the head, indirectly leading to a reduction in the integration rate of the tilt estimator. Much like the direct feedback loop, the indirect feedback loop's influence increases as the separation between the estimated orientation of gravity and the gravito-inertial acceleration signal encoded by the otolith signals increases. The indirect feedback loop has been proposed to account for several perceptual experiences. Two such examples are the illusory tilt experienced immediately following cessation of angular rotation and the continued perception of angular rotation during prolonged constant velocity rotation in the dark (the angular velocity signal from the vestibular semicircular canals will have decayed by this time) [10, 11, 36, 42]. During the presentation of angular visual motion on its own, the lack of accompanying head tilt results in a separation of the estimated orientation of gravity and the otolith signal. The indirect feedback mechanism acts to reduce the angular velocity inferred by the brain thereby reducing the rate of separation between the estimated orientation of gravity and the otolith signal, resulting in a decrease in the rate of bias of the perceived orientation of gravity.

This visual-vestibular processing model can also be conceptualized as a Bayesian model with priors for zero angular velocity and linear acceleration [9, 26, 35, 42]. Such Bayesian priors reflect an adaptation of the brain to the natural statistics of our motion. Namely, since prolonged non-zero linear accelerations and angular velocities are improbable, over time the brain's estimate of angular velocity and linear acceleration will decay towards zero. This latter 'acceleration prior' will cause the motion perceived during prolonged linear acceleration to transition slowly from a perception of linear acceleration to a perception of tilt, due to linear acceleration's yoking to the estimate of gravity (tilt) and tilt being a statistically more probable event than prolonged linear acceleration [9, 26, 42]. "

---

## [Decision Letter · Decision Letter 1]

12 Dec 2019

Variance based weighting of multisensory head rotation signals for verticality perception

PONE-D-19-20795R1

Dear Dr. Dakin,

We are pleased to inform you that your manuscript has been judged scientifically suitable for publication and will be formally accepted for publication once it complies with all outstanding technical requirements.

With kind regards,

Kei Masani

Academic Editor

PLOS ONE

Additional Editor Comments (optional):

Reviewers' comments:

Reviewer's Responses to Questions

**Comments to the Author**

1. If the authors have adequately addressed your comments raised in a previous round of review and you feel that this manuscript is now acceptable for publication, you may indicate that here to bypass the “Comments to the Author” section, enter your conflict of interest statement in the “Confidential to Editor” section, and submit your "Accept" recommendation.

Reviewer #1: All comments have been addressed

Reviewer #2: All comments have been addressed

2. Is the manuscript technically sound, and do the data support the conclusions?

Reviewer #1: Yes

Reviewer #2: (No Response)

3. Has the statistical analysis been performed appropriately and rigorously? 

Reviewer #1: Yes

Reviewer #2: (No Response)

4. Have the authors made all data underlying the findings in their manuscript fully available?

Reviewer #1: Yes

Reviewer #2: (No Response)

5. Is the manuscript presented in an intelligible fashion and written in standard English?

Reviewer #1: Yes

Reviewer #2: (No Response)

6. Review Comments to the Author

Reviewer #1: The authors have addressed all of my concerns. This study is scientifically and technically sound.

Thank you.

Reviewer #2: (No Response)

7. PLOS authors have the option to publish the peer review history of their article (what does this mean?). If published, this will include your full peer review and any attached files.

Reviewer #1: Yes: Jerome Carriot

Reviewer #2: No

---

## [Editor Report · Acceptance letter]

3 Jan 2020

PONE-D-19-20795R1 

Variance based weighting of multisensory head rotation signals for verticality perception 

Dear Dr. Dakin:

I am pleased to inform you that your manuscript has been deemed suitable for publication in PLOS ONE. Congratulations! Your manuscript is now with our production department. 

With kind regards,

on behalf of

Dr. Kei Masani 

Academic Editor

PLOS ONE